# Unveiling New Genetic Variants Associated with Age at Onset in Alzheimer’s Disease and Frontotemporal Lobar Degeneration Due to *C9orf72* Repeat Expansions

**DOI:** 10.3390/ijms25137457

**Published:** 2024-07-07

**Authors:** Antonio Longobardi, Sonia Bellini, Roland Nicsanu, Andrea Pilotto, Andrea Geviti, Alessandro Facconi, Chiara Tolassi, Ilenia Libri, Claudia Saraceno, Silvia Fostinelli, Barbara Borroni, Alessandro Padovani, Giuliano Binetti, Roberta Ghidoni

**Affiliations:** 1Molecular Markers Laboratory, IRCCS Istituto Centro San Giovanni di Dio Fatebenefratelli, 25125 Brescia, Italy; sbellini@fatebenefratelli.eu (S.B.); roland.darazs90@gmail.com (R.N.); csaraceno@fatebenefratelli.eu (C.S.); rghidoni@fatebenefratelli.eu (R.G.); 2Neurology Unit, Department of Clinical and Experimental Sciences, University of Brescia, 25123 Brescia, Italy; pilottoandreae@gmail.com (A.P.); tolassichiara@gmail.com (C.T.); i.libri@unibs.it (I.L.); bborroni@inwind.it (B.B.); alessandro.padovani@unibs.it (A.P.); 3Neurology Unit, Department of Continuity of Care and Frailty, ASST Spedali Civili Hospital, 25123 Brescia, Italy; 4Neurobiorepository and Laboratory of Advanced Biological Markers, University of Brescia and ASST Spedali Civili Hospital, 25123 Brescia, Italy; 5Service of Statistics, IRCCS Istituto Centro San Giovanni di Dio Fatebenefratelli, 25125 Brescia, Italy; ageviti@fatebenefratelli.eu (A.G.); afacconi@fatebenefratelli.eu (A.F.); 6MAC-Memory Clinic and Molecular Markers Laboratory, IRCCS Istituto Centro San Giovanni di Dio Fatebenefratelli, 25125 Brescia, Italy; sfostinelli@fatebenefratelli.eu; 7Cognitive and Behavioural Neurology, ASST Spedali Civili Hospital, 25123 Brescia, Italy; 8Brain Health Center, University of Brescia, 25123 Brescia, Italy

**Keywords:** Alzheimer’s disease, Frontotemporal lobar degeneration, *GRN*, *C9orf72*, age at onset, transferrin, calsyntenin-1, genetic modulators

## Abstract

Alzheimer’s disease (AD) and Frontotemporal lobar degeneration (FTLD) represent the most common forms of neurodegenerative dementias with a highly phenotypic variability. Herein, we investigated the role of genetic variants related to the immune system and inflammation as genetic modulators in AD and related dementias. In patients with sporadic AD/FTLD (n = 300) and *GRN*/*C9orf72* mutation carriers (n = 80), we performed a targeted sequencing of 50 genes belonging to the immune system and inflammation, selected based on their high expression in brain regions and low tolerance to genetic variation. The linear regression analyses revealed two genetic variants: (i) the rs1049296 in the transferrin (*TF*) gene, shown to be significantly associated with age at onset in the sporadic AD group, anticipating the disease onset of 4 years for each SNP allele with respect to the wild-type allele, and (ii) the rs7550295 in the calsyntenin-1 (*CLSTN1*) gene, which was significantly associated with age at onset in the *C9orf72* group, delaying the disease onset of 17 years in patients carrying the SNP allele. In conclusion, our data support the role of genetic variants in iron metabolism (*TF*) and in the modulation of the calcium signalling/axonal anterograde transport of vesicles (*CLSTN1*) as genetic modulators in AD and FTLD due to *C9orf72* expansions.

## 1. Introduction

Alzheimer’s disease (AD) and Frontotemporal lobar degeneration (FTLD) represent the most common forms of neurodegenerative diseases. AD is the most common cause of dementia, accounting for 60% to 80% of cases [1]. AD is mainly characterized by the deposition of beta-amyloid (Aβ) and hyperphosphorylated tau peptides, resulting in neuronal death, inflammation, and atrophy of the brain tissue, which play a crucial role in the onset of symptoms and disease progression [2,3]. FTLD, a clinically heterogeneous neurodegenerative disorder, represents one of the most common causes of early onset dementia, with symptoms often occurring between 45 and 65 years old [4], and is characterized by the accumulation of different proteins, such as microtubule-associated protein tau (MAPT), ubiquitin, TAR DNA-binding protein 43 (TDP-43), and fused in sarcoma (FUS) [5].

Although sporadic forms of FTLD are still poorly understood due to the absence of a clear genetic etiology, 30–50% of patients with FTLD present a positive family history of dementia, with most of them carriers of mutations in genes known to be pathogenic for the disease [6,7,8], such as the chromosome 9 open reading frame (*C9orf72*) [9,10], the granulin precursor (*GRN*) [11,12], and the *MAPT* genes [13,14]. The most common genetic form of FTLD is represented by pathological expansions (>30) of a hexanucleotide repeat (GGGGCC) in the first intron/promoter of the *C9orf72* gene [9,10], leading to a haploinsufficiency due to the reduced expression of C9orf72 and the production of toxic dipeptide repeat protein aggregates [15]. The presence of intermediate expansions (12–30 hexanucleotide repeats) is associated with a risk of developing familial/sporadic FTLD and could influence the clinical phenotypes, including age at onset [16,17]. *GRN* mutations are responsible for up to 25% of familial FTLD cases [18], with the majority represented by loss-of-function mutations causing a reduction in circulating progranulin protein [19]. Progranulin is involved in neuroinflammation and acts as a neuroprotective factor [20]. It is mostly localized in lysosomes, influencing lysosomal acidification and enzymatic activity [21,22,23]. Indeed, FTLD due to *GRN* null mutations or *C9orf72* expansions seem to share molecular and pathological mechanisms, such as a reduction in functional proteins and the presence of lysosomal dysfunction and inflammation [9,19,24,25].

Studies on neurodegenerative diseases, including AD and FTLD, have proposed that a dysregulation of the immune system contributes to neurodegeneration [26,27]. The immune system is deeply involved in the maintenance of tissue homeostasis and injury recovery, acting as a beneficial player. In the presence of a tissue injury or pathogens, inflammatory molecules are transcriptionally induced by the innate immune system to initiate the inflammatory process, which is resolved once the tissue injury has been repaired [28]. Conversely, if the inflammatory stimulus is not resolved, the immune system can be overwhelmed leading to chronic inflammation. Indeed, in neurodegenerative diseases, inflammation has been suggested to be not only a consequence of neurodegeneration but also a crucial player in this process [27]. Moreover, even if AD and FTLD have different pathogenetic mechanisms, they share the hallmarks of neuroinflammation and autoimmunity [29,30,31]. In AD brains and cerebrospinal fluid (CSF), various pro-inflammatory molecules, such as cytokines and complement proteins, are present. Moreover, it was reported that in the proximity of Aβ plaques, microglia cells show more surface proteins such as the major histocompatibility complex class II (MHC-II) which proliferate for the removal of Aβ plaques [32]. Similarly, especially in FTLD due to *GRN* and *C9orf72* mutations, the evidence has shown that altered levels of pro-inflammatory cytokines were present in the brains and serum of patients, with an enhanced inflammation and microglia activation [33,34,35].

Genetic studies further support an involvement of the immune system in neurodegeneration. A large metanalysis suggested the *HLA-DR15* locus, encoding for the major MHC-II protein HLA-DR, as a risk factor for AD [36]; moreover, in a large GWAS of FTLD [37], the *HLA-DRA*/*DRB5* locus was associated with disease risk and, most interestingly, genetic variants in the same locus were demonstrated to influence disease onset in *C9orf72* expansion carriers [38]. The heterogeneity of phenotypic expression in genetic FTLD, even in patients carrying the same mutation, suggests the presence of potential genetic modifiers that influence phenotypic features, such as age at onset, age at death, and disease duration [39]. Accordingly, potential genetic modifiers of age at onset and disease risk were identified through GWAS studies in patients with FTLD carrying *GRN* mutations or *C9orf72* expansions [38,40,41].

In this study, we investigated the role of genetic variants related to the immune system and inflammation as genetic modulators in AD and related dementias. To this aim, we performed targeted sequencing of 50 genes belonging to the immune system and inflammation in a large group of sporadic AD/FTLD patients and *GRN*/*C9orf72* mutation carriers to evaluate the presence of potential genetic modulators associated with the disease phenotype.

## 2. Results

### 2.1. Subjects

A total of n = 380 subjects, comprising n = 150 sporadic AD, n = 150 sporadic FTLD, n = 40 *GRN* mutation carriers (n = 28 genetic FTLD and n = 12 pre-symptomatic subjects), and n = 40 *C9orf72* intermediate/pathological expansion carriers (n = 38 genetic FTLD and n = 2 pre-symptomatic subjects), were screened for the presence of variants in the coding regions of 50 candidate genes belonging to the immune system and inflammation (Table 1). The average age for pre-symptomatic subjects was 52.7 ± 10.1 for *GRN* mutation carriers and 44.0 ± 0.0 for *C9orf72* expansion carriers.

### 2.2. Single Variant Association Study

Considering all the variants detected in the selected 50 genes, we selected those with a Minor Allele Frequency (MAF) greater than 0.01 for the single variant association study in the four diagnostic groups separately (sporadic AD, sporadic FTLD, *GRN*, and *C9orf72*) as well as in the sporadic group (AD + FTLD), in the genetic group (*GRN* + *C9orf72*), and in the whole group (sporadic AD + sporadic FTLD + *GRN* + *C9orf72*). The association between age at onset, diagnostic group, and genetic variants in the immune system and inflammation genes was evaluated by linear regression analysis.

The linear regression analysis revealed, after a 5% false discovery rate (FDR) correction, two genetic variants associated with the age at onset: the rs1049296 (Transferrin gene, *TF*, c.1765C>T, p.Pro589Ser) significantly associated with age at onset in the sporadic AD group and the rs7550295 (Calsyntenin-1 gene, *CLSTN1*, c.994C>T, p.Ala332Thr) significantly associated with age at onset in the *C9orf72* group (Table 2). For the sporadic FTLD and *GRN* groups, no variants were found to be significantly associated with age at onset. The linear regression analyses for genetic variant associations with age at onset were performed excluding *GRN*/*C9orf72* pre-symptomatic carriers.

The chi-squared test was used to evaluate the association of the variants with the diagnostic groups instead and revealed no significant associations.

### 2.3. The rs1049296 TF Variant Is Associated with Sporadic AD

In the sporadic AD group, the rs1049296 *TF* variant led to a decrease in the average age at onset from 74 years for the homozygous wild-type allele (C/C, n = 101) to an age of 69 years for the heterozygous SNP allele (C/T, n = 38) and 65 years for the homozygous SNP allele (T/T, n = 4) (Figure 1).

This suggests that, with respect to the wild-type allele, each SNP allele could reduce the age at onset by 4.34 years (p_adj_ = 0.010). Accordingly, the COX regression also showed that the rs1049296 *TF* variant is significantly associated with the age at onset (p_adj_ = 0.039), suggesting that each SNP allele could increase the risk of developing the disease by 1.72 times (95% CI: 1.24 ÷ 2.37). The dominant model confirms the association of the rs1049296 *TF* with the age at onset, suggesting a reduction in the age at onset of 4.68 years (p_adj_ = 0.022). No associations were found in the recessive model.

The rs1049296 *TF* genotypes are equally distributed according to sex in the sporadic AD group (*p* = 0.40). The association of the rs1049296 *TF* variant with the age at onset is also significant in the sporadic group (AD + FTLD) for both the additive and the dominant model (p_adj_ = 0.017 and p_adj_ = 0.005, respectively), but not in the sporadic FTLD group alone. The rs1049296 *TF* genotypes are equally distributed according to sporadic AD/FTLD group (*p* = 0.87) and sex (*p* = 0.51). Moreover, considering the whole group (sporadic AD + sporadic FTLD + *GRN* + *C9orf72*), the rs1049296 *TF* variant was nominally associated with age at onset (*p*-value not significant after FDR correction).

Furthermore, to determine the median age at onset for the different rs1049296 *TF* genotypes observed in the sporadic AD group, we employed the Kaplan–Meier estimate (Figure 2). The median age at onset was 75 years (95% CI: 73–76) for homozygous wild type C/C carriers and significantly lower for heterozygous C/T carriers (71 years, 95% CI: 69–73) and for homozygous T/T carriers (70.5 years, 95% CI: 45 to NA *; * the number of observations was too small to estimate the upper limit of the confidence interval) (*p* log-rank test = 0.0034).

### 2.4. The rs7550295 CLSTN1 Variant Is Associated with C9orf72

In the *C9orf72* group, the rs7550295 *CLSTN1* variant led to an increase in the average age at onset from 62 years for the homozygous wild-type allele (C/C, n = 32) to an age of 79 years for the heterozygous SNP allele (C/T, n = 3) (Figure 3).

This variant was associated with a delay in the age at onset of 17.13 years with respect to the wild-type allele (p_adj_ = 0.006). However, due to the limited number of heterozygous subjects (n = 3), these results should be interpreted with caution. The rs7550295 *CLSTN1* genotypes are equally distributed according to sex in the *C9orf72* group (*p* = 0.54). The association of the rs7550295 *CLSTN1* variant with the age at onset is also significant in the genetic group (*GRN* + *C9orf72*) (p_adj_ = 0.021), but not in the *GRN* group alone. The rs7550295 *CLSTN1* genotypes are equally distributed according to *GRN*/*C9orf72* mutation (*p* = 0.62) and sex (*p* = 0.62).

### 2.5. Variants Interpretation

The identified *TF* and *CLSTN1* missense variants were interpreted using automatic and manual annotations from various computational tools or databases (Table 3).

The rs1049296 *TF* variant showed a frequency of 16/16% (Genome/Exome) in the Non-Finnish European (NFE) population and of 14/16% (Genome/Exome) in the whole population, according to the gnomAD datasets. Other frequencies of the rs1049296 *TF* variant in various populations are listed in Appendix A. The variant has been reported to be a tolerated polymorphism by many in silico prediction tools, including CADD, Polyphen-2, SIFT, FATHMM, and Mutation Taster. Additionally, the fact that the variant site reported a negative score for GERP suggests that it follows the neutral rate of evolution and should have no damaging effects. According to MUPro and I-Mutant, the variant is predicted to decrease TF protein stability (ΔΔG = −0.801 and ΔΔG = −1.65, respectively), while in Missense3D-DB it is predicted to be structurally neutral. In the public database ClinVar, the rs1049296 *TF* variant was described as being benign in congenital hypotransferrinemia conditions and associated with AD as a risk factor [42,43]. The Human Gene Mutation Database (HGMD) also confirmed its association with AD.

The rs7550295 *CLSTN1* variant showed a frequency of 5/5% (Genome/Exome) in the NFE population and of 12/8% (Genome/Exome) in the whole population. Other frequencies of the rs7550295 *CLSTN1* variant in various populations are listed in Appendix A. The variant was predicted to be benign/tolerated by the previously mentioned in silico tools, although it was not reported in ClinVar, nor in HGMD. Moreover, according to MUPro and I-Mutant, the variant should decrease CLSTN1 protein stability (ΔΔG = −0.474 and ΔΔG = −0.83, respectively).

## 3. Discussion

Neurodegenerative dementias, particularly the genetic forms of these diseases, can show a high phenotypic variation with regard to presenting symptoms, disease course, and age at onset. The mechanisms underlying phenotypic variations in neurodegenerative dementias are beginning to be elucidated.

Several studies have demonstrated the presence of genetic loci involved in the immune system and inflammation that represent risk factors and/or could influence age at onset in neurodegenerative diseases. More specifically, (i) a large GWAS study demonstrated an association between FTD and the *HLA* locus (6p21.3), encoding proteins which play a pivotal role in the antigen presentation of intracellular and extracellular peptides and in the regulation of innate and adaptive immune responses [37]; (ii) the presence of a variant in the *C6orf10/LOC101929163* locus was reported to influence brain expression of *HLA-DRB1* and to be associated with age at onset in *C9orf72* expansion carriers [38]; (iii) a large meta-analysis suggested the *HLA-DR15* locus, a locus encoding for the major MHC II protein HLA-DR, as a risk factor for AD, suggesting an involvement of the immune system process in this disease [36].

The aim of our study was to identify a genetic profile of the immune system associated with disease onset and clinical phenotype in a group of patients with neurodegenerative dementias. Understanding the links between genetic variation and phenotypic features in patients with different forms of neurodegenerative dementias could help us to decipher the underlying mechanisms, thus allowing us to gain a deeper understanding of the clinical observations. To this aim we performed a target sequencing of 50 genes involved in the immune system and inflammation, highly expressed in brain regions of interest and with a high intolerance to variation, in a large group of patients with sporadic AD and FTLD, and in a subgroup of genetic FTLD cases carrying *GRN*/*C9orf72* mutations. We selected common variants in our dataset with an MAF > 0.01 to perform association studies with age at onset in the single/combined patients and the whole group. Two variants were found to be associated with the age at onset, specifically, (i) the *TF* p.Pro589Ser (rs1049296) in the sporadic AD group, anticipating the disease onset of 4 years in patients carrying the homozygous allele (T/T) with respect to the heterozygous allele (C/T) and of 9 years with respect to the wild-type allele (C/C); (ii) the *CLSTN1* p.Ala332Thr (rs7550295) in the *C9orf72* group, delaying the onset of 17 years in patients carrying the heterozygous allele (C/T) with respect to the wild-type allele (C/C), suggesting a potential protective effect of the polymorphism in the presence of *C9orf72* expansions.

Transferrin, encoded by *TF* gene, is an iron-binding glycoprotein synthesized by the liver with a central role in iron transport through receptor-mediated endocytosis which also acts as a negative acute-phase protein which decreases during inflammation [44,45,46]. The *TF* gene is characterized by a significant degree of genetic polymorphisms with rs1049296 *TF*, namely *TF* C2 as the most common variant [47,48]. The *TF* C2 variant is reported to be a risk factor for AD, in synergy with the p.Cys282Tyr (rs1800562) allele of the Homeostatic Iron Regulator (*HFE*) gene: the combination of *TF* C2 and the *HFE* p.Cys282Tyr might lead to an excess of redox-active iron and eventually the generation of oxidative stress in the preclinical phase of AD [48,49,50]. Moreover, it was shown that *TF* may be involved in limiting the amyloid aggregation process, as suggested by the association found in the CSF of AD patients between increased Aβ42/Aβ40 ratio and the presence of the rs1049296 *TF* variant [51].

The presence of the rs1049296 *TF* homozygous/heterozygous allele which is associated with an earlier onset compared to the *TF* wild-type allele, as shown in our study, suggests a potential altered function of the transferrin protein, and therefore, an alteration to iron homeostasis that could be implicated in the pathogenesis of the disease. Accordingly, several studies have revealed a relationship between iron and AD, in particular the increased risk of AD related to the dysregulation of iron metabolism, as recently reviewed in [52]. 

Of note, the rs1049296 *TF* variant has also been found to be significantly associated with age at onset in the whole sporadic group (AD + FTLD) and only nominally associated in the whole group (sporadic AD + sporadic FTLD + *GRN* + *C9orf72*). The lack of an association in the different groups separately (sporadic FTLD, *GRN*, and *C9orf72*) suggests that the associations in the combined groups are driven by the association of the sporadic AD group rs1049296 *TF*, which is in line with the results reported in the literature.

Encoded by the *CLSTN1* gene, the calsyntenin-1 protein, also referred as Alcadeinα, is a transmembrane protein of the postsynaptic membrane, abundant in most neurons of the central nervous system (CNS). It is deeply involved in the modulation of calcium signalling in the postsynaptic membrane and, after its internalization due to extracellular proteolytic cleavage, in the intracellular Ca^2+^ reserves [53].

Calsyntenin-1 also has a pivotal role in the axonal anterograde transport of vesicles. It has been demonstrated that calsyntenin-1 can induce a vesicle association with the kinesin-1 motor for axonal transport of cargo, competing with the transport of APP-containing vesicles [54]. On the other hand, calsyntenin-1 and APP have been found to be colocalized in the dystrophic neurites and senile plaques of AD brain specimens, and furthermore, some calsyntenin-1-containing vesicles also contain the APP protein [55,56]. Accordingly, calsyntenin-1 was found to be reduced in AD brains and correlated with increased levels of Aβ, suggesting that an interruption of the calsyntenin-1-associated axonal transport of APP could be a pathogenic mechanism in AD, leading to an increased production of Aβ [57,58].

Several studies on the calsyntenin-1 CSF levels reported an association of the protein with FTLD; specifically, it was found that calsyntenin-1 CSF levels were lower in patients affected by FTLD compared to AD and cognitively normal controls [59,60,61]. Moreover, the combination of calsyntenin-1 with other synaptic proteins has shown a potential ability to discriminate FTLD subtypes from other type of dementias, such as FTLD TDP-43-subtype from AD and healthy subjects, and *GRN*/*C9orf72* pre-symptomatic mutation carriers from mutation non-carriers [61,62].

In our study, the heterozygous rs7550295 *CLSTN1* variant was found to be associated with the age at onset in the *C9orf72* group, leading to a delay in the age at onset compared to the wild-type allele. This suggests that the presence of the SNP could act as a potential protective genetic modulator. Interestingly, recent studies have described a pathogenic link between *C9orf72* expansions and the dysregulation of calcium signalling [63,64]. Since the *CLSTN1* gene is involved in the modulation of calcium signalling, this could explain the modulation of the *CLSTN1* variant of the phenotypic trait of *C9orf72* expansion carriers.

The lack of genetic variants associated with the diagnostic group, especially in the genetic groups, could be due to the limited number of samples, representing a limitation of the study. Therefore, our results should be interpreted with caution, especially for the small number of rs7550295 *CLSTN1* heterozygous variant carriers and the lack of homozygous carriers. Future replication analyses on a larger number of samples may be useful for confirming and strengthening our results.

In conclusion, our data support a role of genetic variants related to immune system and inflammation as genetic modulators in neurodegenerative dementias. These genes are involved in the iron metabolism (*TF*) and in the modulation of calcium signalling at the postsynaptic level as well as in the axonal anterograde transport of vesicles (*CLSTN1*). Of note, the data arising from this study suggest that genetic modulators are disease specific. Thus, the stratification of patients according to modifying factors might also be incorporated into clinical trials in the near future.

## 4. Materials and Methods

### 4.1. Participants

This retrospective study was carried out on DNA from a total of n = 380 subjects, comprising n = 150 sporadic AD, n = 150 sporadic FTLD, n = 40 *GRN* mutation carriers (n = 28 genetic FTLD and n = 12 pre-symptomatic subjects), and n = 40 *C9orf72* intermediate/pathological expansion carriers (n = 38 genetic FTLD and n = 2 pre-symptomatic subjects). The genetic features, including *GRN* mutations, *C9orf72* repeat expansions, and *APOE* genotype (when available) are listed in Appendix A. Clinical diagnoses of AD and FTLD were made according to international guidelines [65,66,67,68,69]. DNA samples were available from Biobanca IRCCS Centro San Giovanni di Dio Fatebenefratelli, Brescia (BioBank FBF; bbmri-eric ID: IT_138442378660827 and Orphanet Biobank) and NeuroBiorepository of ASST Spedali Civili Brescia. *C9orf72* and *GRN* genetic screening was previously performed as described in [17,18,70,71]. Written informed consent was obtained from all subjects. The study protocol was approved by the local ethics committees (Prot. N. 79/2020, date of approval 21 December 2020; Prot NP 1471, DMA, Brescia, approved in its last version on 20 December 2020).

### 4.2. Gene Selection

Panel genes were chosen from gene sets downloaded from 5 different databases: NCBI Gene [https://www.ncbi.nlm.nih.gov/gene/], GSEA [https://www.gsea--msigdb.org/gsea/index.jsp], Kegg [https://www.genome.jp/kegg/], Gene Ontology [http://geneontology.org/], and Reactome [https://reactome.org/], all accessed on 5 April 2022. Gene selection was performed using search terms linked to the immune system. Four terms in particular were used: “Innate Immune System”, “Adaptive Immune System”, “Inflammation”, and “Autoimmune”. Gene sets were then merged and duplicates were removed. A list of 13,276 unique genes was obtained. All the genes were then filtered depending on their expression levels in multiple brain regions (i.e., amygdala, caudate basal ganglia, cortex, frontal cortex, hippocampus, and putamen basal ganglia). Gene expression levels were downloaded from the GTEx portal [https://gtexportal.org/home/ (accessed on 5 April 2022)]. Expression percentiles were then calculated, and all the genes that fell above the 95th percentile for all the brain regions of interest were selected. From such selection a list of 476 genes was obtained, which was ordered based on genes Residual Variation Intolerance Score (RVIS) (ascending order). The first 50 genes that fell below the 25th percentile of the RVIS were then selected for the gene panel design (Appendix A).

### 4.3. Genetic Analyses

The entire coding region of the 50 candidate genes were analyzed by amplicon-based target enrichment and Next-Generation Sequencing (NGS) of the exons and exon-intron boundaries on Illumina NextSeq2000 system (Illumina, San Diego, CA, USA). The quality assessment of gDNA was performed on a 0.8% agarose gel, and gDNA was quantified with a Qubit dsDNA BR Assay Kit (Thermo Fisher Scientific, Waltham, MA, USA). A total of 200 ng of gDNA was used for library preparation with Illumina DNA Prep with Enrichment kit (Illumina, San Diego, CA, USA). gDNA was tagmented, amplified, and purified with Illumina Purification Beads (Illumina, San Diego, CA, USA). The size, quality, and quantity of libraries were assessed with a High Sensitivity DNA kit on a Bioanalyzer instrument (Agilent Technologies, Santa Clara, CA, USA). A 1000 pM sample of the pooled library was loaded with NextSeq 1000/2000 P1 reagents (Illumina, San Diego, CA, USA) and sequenced on an Illumina NextSeq 2000 system (Illumina, San Diego, CA, USA).

### 4.4. Bioinformatics Analysis: Data Pre-Processing, Mapping, and Variant Calling

The quality assessment of the sequenced reads was conducted employing FastQC (version 0.11.9) [http://www.bioinformatics.babraham.ac.uk/projects/fastqc/], accessed on 15 April 2024. Subsequently, Trimmomatic (version 0.39) [72] was utilized to eliminate adapters and reads of substandard quality. The resultant high-quality reads were aligned against the reference genome (hg19) via the bwa-mem aligner (0.7.17-r1188) [73]. A thorough coverage analysis ensued through the DepthOfCoverage module of the Genome Analysis Toolkit (GATK, v4.3.0.0) [https://gatk.broadinstitute.org/]. Notably, all samples exhibited a coverage of no less than 40X across all 50 panel genes. Following alignment and coverage assessment, duplicated reads were identified and marked using Picard’s MarkDuplicates module (version 2.27.5) [https://broadinstitute.github.io/picard/]. Afterwards, the quality of every single base call was evaluated and recalibrated through BQSR (Base Quality Score Recalibration) in order to tackle systematic technical errors. The BaseRecalibrator and the ApplyBQSR modules of GATK (v4.3.0.0) were used for this task. Finally, SNVs (Single-Nucleotide Variants) and INDELs (Insertions/Deletions) calling was executed employing the HaplotypeCaller module of GATK (v4.3.0.0), with the Single-Nucleotide Polymorphism Database (dbSNP; v150) that served as the reference database for the variants. To ensure the reliability of identified variants, stringent filtering criteria were applied in accordance with GATK hard-filtering guidelines. SNVs that met the following criteria were excluded: QUAL (Quality) < 30, DP (Site Depth) < 20, QD (Quality by Depth) < 2, MQ (Root Mean Square Mapping Quality) < 40, FS (Fisher Strand test) > 60, SOR (Strand Odds Ratio test) > 3, MQRankSum (Rank Sum test for Mapping Quality) < −12.5, ReadPosRankSum (Rank Sum test for Site Position) < −8.0, Genotype GQ (Genotype Quality) < 20, and Genotype DP (Genotype Depth) < 20.0. INDELs that met the following criteria were excluded: QUAL (Quality) < 30, QD (Quality by Depth) < 2, FS (Fisher Strand test) > 200, ReadPosRankSum (Rank Sum test for Site Position) < −20.0, Genotype GQ (Genotype Quality) < 20 and Genotype DP (Genotype Depth) < 20. Furthermore, both SNVs and INDELs with an allele balance < 25% were discarded. At last, the functional impact of each variant was evaluated through ANNOVAR (version 2020-06-08) [74]. Among the many functional annotations applied by ANNOVAR (being manually curated or from computational tools), the following were of particular interest for the aim of this study: ClinVar [https://www.ncbi.nlm.nih.gov/clinvar/] [75], CADD (v1.7) [https://cadd.gs.washington.edu/] [76], Polyphen-2 (version 2.2.3) [http://genetics.bwh.harvard.edu/pph2/index.shtml] [77], Sift [https://sift.bii.a--star.edu.sg/] [78], FATHMM (v2.3) [http://fathmm.biocompute.org.uk/] [79], Mutation Taster [https://www.mutationtaster.org/] [80], and GERP [http://mendel.stanford.edu/SidowLab/downloads/gerp/] [81], all accessed on 2 May 2024.

Population frequencies of variants were obtained from the gnomAD (2.1.1) [https://gnomad.broadinstitute.org/ (accessed on 27 June 2024)] [82] database. Additional annotations were retrieved from OMIM [https://www.omim.org/], and HGMD [https://www.hgmd.cf.ac.uk/ac/index.php] [83], accessed on 2 May 2024. Protein stability predictions were performed through I-Mutant (v2.0) [https://folding.biofold.org/i--mutant/i--mutant2.0.html] [84], MuPro (v1.0) [http://mupro.proteomics.ics.uci.edu/] [85], and Missense3D-DB (v.1.5.4) [http://missense3d.bc.ic.ac.uk:8080/home] [86], all accessed on 21 May 2024.

### 4.5. Statistical Analysis

The age at evaluation and age at onset were reported as mean and standard deviation, while categorical variables, as sex, were presented as numbers and percentages. The normality of continuous features was assessed using the Shapiro–Wilk test and graphical inspection. Kruskal–Wallis test with Dunn’s post hoc tests was used for the group comparisons of non-normally distributed variables. Chi-squared test was used to analyze differences in sex distribution among the study groups.

To identify genetic variants with allele frequencies that systematically vary as a function of age at onset, we performed a single variant association analysis, excluding *GRN*/*C9orf72* pre-symptomatic carriers. Additive, dominant, and recessive models were implemented for each diagnostic group to identify variants associated with the age at onset for specific diagnoses. We focused on non-synonymous variants (missense, splicing, stop-gain, stop-loss, conservative, or frameshift ins/del variants) that were either low frequency (0.01 < MAF < 0.05) or common (MAF > 0.05) in our dataset. We modelled the quantitative trend in onset using linear regression analysis. The estimates were adjusted for sex, and *p*-values were corrected using the FDR (Benjamani–Hochberg method with a 5% threshold). A Cox proportional hazard regression model was used to assess the potential risk associated with each allele of the selected variants, censoring pre-symptomatic subjects at their age at evaluation. The estimates were adjusted for sex. The different incidence rates among genotypes of significant variants were illustrated using Kaplan–Meier curves, including both patients and pre-symptomatic subjects where applicable, with censoring at age at evaluation. Additionally, a chi-squared test was performed on the entire sample to test the association between the presence of specific variants and the diagnosis, with *p*-values corrected using FDR (Benjamani–Hochberg method with a 5% threshold). In the diagnostic groups where more than one subject per family was present, only one patient for each genealogically unrelated pedigree was considered for the analysis. All analyses were conducted using R software (version 4.3.2).

## Figures and Tables

**Figure 1 ijms-25-07457-f001:**
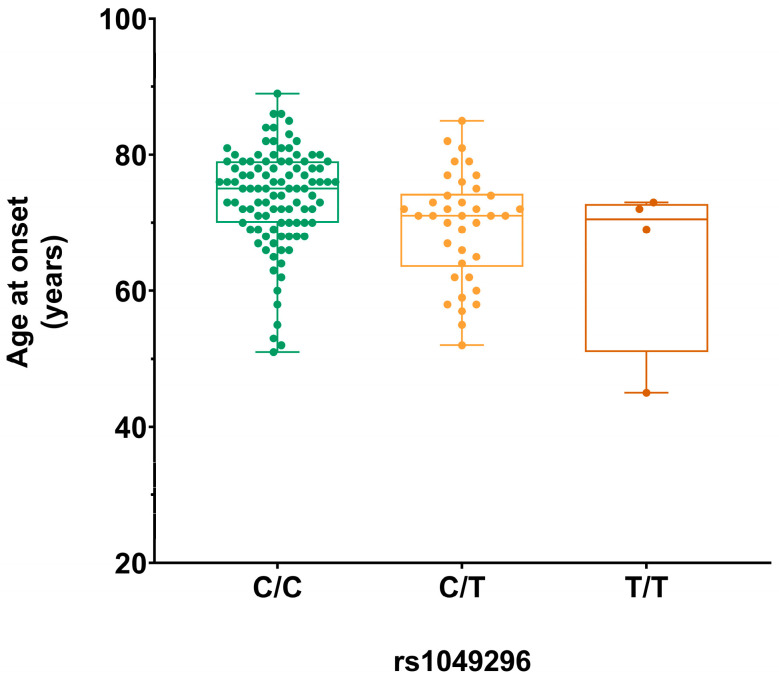
Boxplot of the age at onset distribution according to rs1049296 *TF* genotypes in the sporadic AD group. Homozygous wild-type (C/C), heterozygous (C/T), and homozygous SNP allele (T/T).

**Figure 2 ijms-25-07457-f002:**
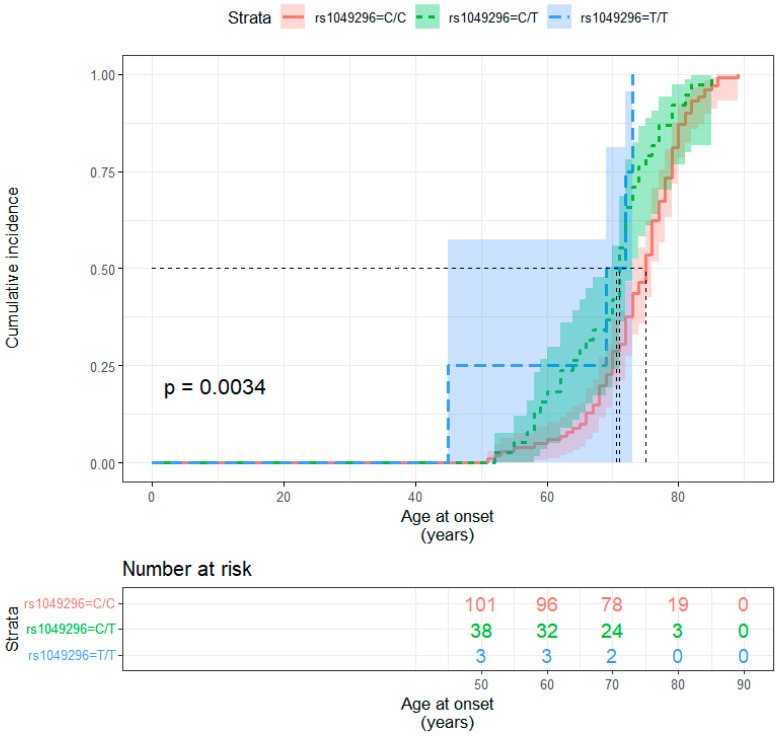
Kaplan–Meier curve showing disease incidence in the three rs1049296 *TF* genotypes.

**Figure 3 ijms-25-07457-f003:**
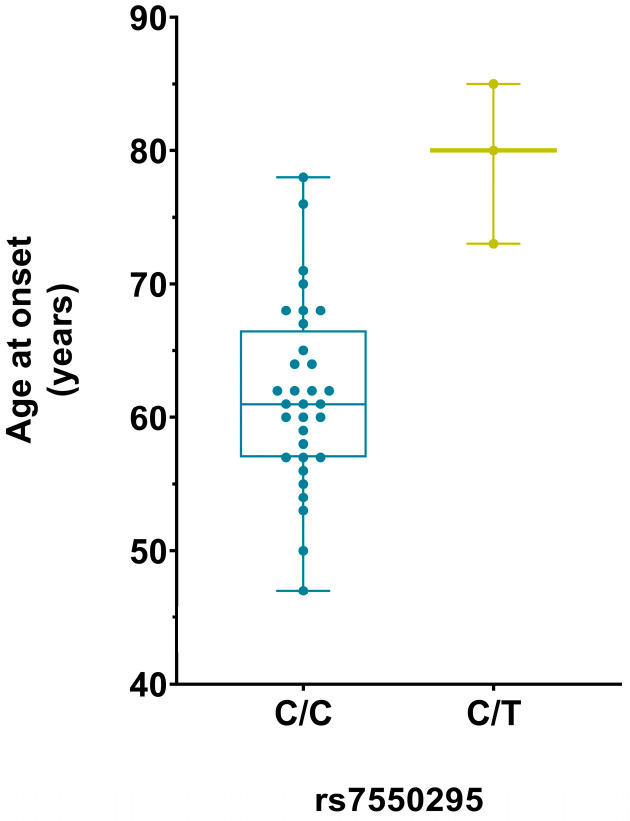
Boxplot of the age at onset distribution according to rs7550295 *CLSTN1* genotypes in the *C9orf72* group. Homozygous wild-type (C/C) and heterozygous SNP allele (C/T).

**Table 1 ijms-25-07457-t001:** Clinical and demographic characteristics of patients included in the study.

	Sporadic AD (n = 150)	Sporadic FTLD (n = 150)	Genetic FTLD		*p* Value
			*GRN* (n = 28)	*C9orf72* (n = 38)	
Age, years	75.7 ± 7.9 *	70.1 ± 8.7 ^#^	63.3 ± 9.2	66.4 ± 8.0	<0.0001 ^a^
Age at onset, years	72.4 ± 7.9 *	66.3 ± 9.2	61.0 ± 9.4	62.8 ± 8.5	<0.0001 ^a^
Sex, % female	69.3% ^$^	44.7%	57.1%	34.2%	<0.0001 ^b^

AD, Alzheimer’s disease patients; FTLD, frontotemporal lobar degeneration patients; *GRN*, *GRN* mutation carriers; *C9orf72*, *C9orf72* intermediate/pathological expansion carriers. Mean ± Standard Deviation. ^a^ Kruskal–Wallis test with Dunn’s post hoc tests; ^b^ chi-squared test; * Sporadic AD vs. sporadic FTLD, *GRN*, and *C9orf72*, *p* value < 0.0001; ^#^ Sporadic FTLD vs. *GRN*, *p* value < 0.05; ^$^ AD vs. FTLD, *C9orf72*, *p* value < 0.0001.

**Table 2 ijms-25-07457-t002:** Variants associated with age at onset.

SNP	Gene	Location	c.pos	p.pos	Group	P Linear	P Linear FDR	Beta Linear	CI (95%) Linear
rs1049296	*TF*	missense	c.1765C>T	p.Pro589Ser	sporadic AD	0.0005	0.010	−4.34	−6.74 ÷ −1.94
rs7550295	*CLSTN1*	missense	c.994C>T	p.Ala332Thr	*C9orf72*	0.0003	0.006	17.13	8.46 ÷ 25.81

*TF*, transferrin; *CLSTN1*, calsyntenin-1; SNP, single nucleotide polymorphism; c.pos, coding position; p.pos, protein position; P Linear, *p* value derived from linear regression; P Linear FDR, *p* value derived from linear regression and corrected with false discovery rate 5%; Beta Linear, beta coefficient derived from linear regression; CI (95%) Linear, 95% confidence interval derived from linear regression.

**Table 3 ijms-25-07457-t003:** *TF* and *CLSTN1* missense variants.

SNP	Gene	Variant	gnomAD_NFE (Genome/Exome)	CADD	Poly-phen2	GERP	FATHMM	SIFT	Mutation Taster	ΔΔG MUPro and I-Mutant
rs1049296	*TF*	p.Pro589Ser	0.159/0.163	0.02	B	−6.9	T	T	P	−0.801/−1.65
rs7550295	*CLSTN1*	p.Ala332Thr	0.049/0.049	0.15	B	−6.0	T	T	P	−0.474/−0.83

gnomAD_NFE, genome aggregation database non-Finnish European; CADD, combined annotation dependent depletion; Poly-Phen2, polymorphism phenotyping v2; B, benign; GERP, genomic evolutionary rate profiling; FATHMM, functional analysis through hidden Markov models; T, tolerated; SIFT, sorting intolerant from tolerant; P, polymorphism automatic; ΔΔG, protein stability free-energy change.

## Data Availability

The data presented in this study are available in the Zenodo Data Repository at doi: 10.5281/zenodo.11659282 [87].

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
