# Peer review of "Unveiling New Genetic Variants Associated with Age at Onset in Alzheimer’s Disease and Frontotemporal Lobar Degeneration Due to C9orf72 Repeat Expansions"

_ijms, 2024, doi:10.3390/ijms25137457_

Round 1

Reviewer 1 Report

Comments and Suggestions for Authors

My suggestions, and questions:

1. What kind of GRN mutations were observed in the carriers? In the case of C9orf72 repeat expansion carriers, how many G4C2 repeats were found in them?

2. Authors may add the APOE E4 allele ratio in Table 1. 

3. Were any c9orf72 repeat expansion or GRN mutations found in AD patients? How about APP, PSEN1, or PSEN2 variants? The authors may include it in a supplement file.

4. Besides GRN and c9orf72, were any pathogenic GRN mutations found in the patients (such as MAPT, TARDBP, CHMP2B, etc). The authors may include it in a supplement file.

5. Is there any possible common pathway between TF, CLSTN1 and c9orf72?

6. What was the frequency of TF and CLSTN1 variants in other populations besides NFE?

7. Do these variants impact the severity of the disease? Especially in the homozygous stage? 

8. Is it possible that TF and CLSTN1 may work as risk modifiers in AD patients too? 

Reviewer 2 Report

Comments and Suggestions for Authors

The article titled "Unveiling new genetic variants associated with age at onset in Alzheimer’s Disease patients and C9orf72 repeat expansion carriers" investigates the role of genetic variants related to the immune system and inflammation as genetic modulators in Alzheimer's disease (AD) and related dementias, specifically Frontotemporal lobar degeneration (FTLD). The article needs revisions before acceptance. Here are some questions and suggestions for improvement:

1. Should “Alzheimer’s Disease” in the title be written as “Alzheimer’s disease” to maintain consistency with other parts of this article?

2. Given that the study also involves Frontotemporal lobar degeneration (FTLD), should the title reflect this to provide a clearer scope of the research? For example, revise the title to include FTLD.

3. The p-values in Table 1 are unclear regarding which groups or comparisons they pertain to. Could the authors clarify the specific comparisons or groups for these p-values?

4. The sample sizes for certain groups in Figures 1 and 3 (e.g., rs1049296 T/T group and rs7550295 C/T group) are quite small. Does this affect the reliability of the conclusions drawn? The authors should discuss the limitations related to small sample sizes in the relevant sections of the discussion to provide a balanced interpretation of the results.

5. Lines 141-143: It is recommended to introduce the full names of all abbreviations first. Then explain "P, Beta, and CI Linear, p-value, beta coefficient, and confidence interval derived from linear regression". Moreover, it is suggested that the terms and phrases in the sentence "P, Beta, and CI Linear, p-value, beta coefficient, and confidence interval derived from linear regression" should be consistent with those in the table.

Round 2

Reviewer 1 Report

Comments and Suggestions for Authors

The authors fulfilled the revision request. Thank you.